# Rational Identification of Ritonavir as IL-20 Receptor A Ligand Endowed with Antiproliferative Properties in Breast Cancer Cells

**DOI:** 10.3390/ijms26031285

**Published:** 2025-02-02

**Authors:** Valentina Maggisano, Adriana Gargano, Jessica Maiuolo, Francesco Ortuso, Francesca De Amicis, Stefano Alcaro, Stefania Bulotta

**Affiliations:** 1Dipartimento di Scienze della Salute, Università degli Studi “Magna Græcia” di Catanzaro, Campus Universitario “S. Venuta”, Viale Europa, 88100 Catanzaro, Italy; vmaggisano@unicz.it (V.M.); a.gargano@unicz.it (A.G.); maiuolo@unicz.it (J.M.); ortuso@unicz.it (F.O.); bulotta@unicz.it (S.B.); 2Associazione CRISEA—Centro di Ricerca e Servizi Avanzati per l’Innovazione Rurale, Località Condoleo, 88055 Belcastro, Italy; 3Net4Science Academic Spinoff, Università “Magna Græcia” di Catanzaro, Campus Universitario “S. Venuta”, Viale Europa, 88100 Catanzaro, Italy; 4Dipartimento di Farmacia e Scienze della Salute e della Nutrizione, Università della Calabria, 87036 Rende, Italy; francesca.deamicis@unical.it; 5Centro Sanitario, Università della Calabria, 87036 Rende, Italy

**Keywords:** virtual screening, DrugBank, ZINC15, TME, IL-20RA, ritonavir, TNBC

## Abstract

Targeting the tumor microenvironment (TME) is an attractive strategy for developing new drugs with anticancer activity against triple-negative breast cancer (TNBC). Interleukins (ILs) are key players in the TME cytokine network promoting cancer progression. Recent studies have highlighted the involvement of IL-20 receptor subunit alpha (IL-20RA) signalling in several cancers, including BC, in which IL-20RA is highly expressed, correlating with poor prognosis and influencing tumoral characteristics such as proliferation, cell death, invasiveness, and TME activity. Therefore, elucidating the role of the IL-20RA signalling pathway could form the basis for developing new therapeutic strategies. This study aimed to identify selective bioactive ligands able to affect IL-20RA activity. Virtual screening of over 310,000 compounds from both the DrugBank and ZINC15 databases identified four potential hit compounds tested for their anticancer activity against TNBC in vitro cell lines. Notably, Ritonavir, a well-known Human Immunodeficiency Virus Type 1 (HIV-1) protease inhibitor, significantly inhibited cell proliferation (about 40% at 50 µM, *p* < 0.001). IL-20 preincubation counteracted Ritonavir’s cytostatic effect while IL-20RA knockdown restored proliferation in Ritonavir-treated TNBC cells. In conclusion, these findings demonstrated that Ritonavir reduced TNBC cell proliferation through IL-20RA activity modulation, suggesting its potential repurposing as a therapeutic agent for TNBC management.

## 1. Introduction

Cancer is a major societal, public health, and economic problem in the 21st century that caused 20 million new cases in the year 2022 alongside 9.7 million deaths [1]. Breast cancer (BC) is the most frequently diagnosed cancer in women globally and the leading cause of cancer death worldwide, in 157 countries for incidence and 112 for mortality [1,2]. BC is a highly heterogeneous solid tumor with several subtypes that differ in morphology, genetics, and clinical behavior [3,4]. To date, locoregional (surgery and radiotherapy) and systemic (chemotherapy, endocrine, and biological therapy) approaches represent the main therapeutic options for the treatment of BC. However, due to the histological heterogeneity and complex molecular profile of the tumor, patients frequently show different responses to the above treatments, suggesting that new insights are needed as a step toward precision medicine [5]. Triple-negative BC (TNBC) represents the more aggressive subtype accounting for 15–20% of all BCs, the vast majority of which are extremely invasive with higher distant metastasis rates, poor prognosis, and short survival. TNBC is characterized by a lack of expression of estrogen receptor (ER), progesterone receptor (PR), and human epidermal growth factor receptor 2 (HER2) (also known as ErbB2). Due to these molecular features, the identification of new prognostic factors and pharmacological tools is needed to improve TNBC management.

An attractive promising approach is based on the use of compounds able to target the components of the tumor microenvironment (TME) [6]. The TME consists of heterogeneous populations including cancer cells themselves, immune cells, fibroblasts, mesenchymal cells and adipocytes, non-cellular elements of the extracellular matrix (ECM), and a plethora of cytokines. In addition, physical properties such as pH and oxygen content influence TME activity. In this context, immune surveillance is responsible for recognizing and eliminating the vast majority of cancer cells. Nevertheless, cancer cells employ various strategies to circumvent this vigilance, an ability recognized as one of the hallmarks of cancer [7]. The impact of the immune environment on cancers is particularly relevant in TNBC as evidenced by more recent works [8,9,10] and, in this scenario, a key role is also played by several interleukins (ILs) that nurture cancer progression [11,12].

In recent decades, increasing evidence has related the involvement of the IL-20 receptor subunit alpha (IL-20RA) axis with the development and progression of many disorders, including inflammatory diseases and cancer [13,14,15]. IL-20RA belongs to IL-20R type 1 trans-membrane receptors forming a heterodimer with IL-20R subunit beta (IL-20RB) to create a functional receptor for IL-20 and its two closest homologues IL-19 and IL-24 [16,17]. IL-20RA, IL-20RB, and their ligands are expressed in a few normal tissues such as skin, and primarily in monocytes of peripheral blood, performing their action through the Janus kinase (JAK)–signal transducer and activator of transcription (STAT) pathway [11]. High levels of IL-20RA expression as well as of IL-19, IL-20, and IL-26 have been observed in several tumors of epithelial origin, including among others prostate, bladder, and breast and gastric cancer, and correlated with poor clinical outcome [18,19,20,21,22]. On the contrary, IL-24 is the only cytokine of the IL-20 subfamily with tumor suppressor activity [23]. Recent data have associated IL-20RA signalling with hallmarks of cancer, including regulation of proliferative signalling, resistance to cell death, and activation of cellular migration and invasion [24,25,26]. Upon ligand binding, IL-20R type 1 phosphorylates, activating JAK that, in turn, phosphorylates STAT. Finally, STAT dimerizes and translocates into the nucleus, regulating the expression of genes involved in cancer progression [14,17,24,25,26]. Notwithstanding, the mechanisms through which IL-20RA regulates TME elements in BC need to be further investigated.

In recent years, in silico approaches have emerged as powerful tools in the field of drug discovery, streamlining the identification and optimization of novel therapeutic compounds. By rapidly evaluating vast chemical libraries, they enable researchers to identify compounds with high binding affinities to target proteins, significantly reducing the time and costs associated with experimental screening. Virtual screening is particularly advantageous when applied to different libraries, such as databases of natural compounds or those of FDA-approved drugs, as it not only exploits the therapeutic potential of bioactive molecules derived from nature but also repurposes clinically validated drugs for new indications.

The potential of in silico techniques extends beyond efficiency; they also provide insights into the molecular mechanisms underpinning drug–target interactions. Techniques like molecular docking, which allows for the prediction of how small molecules interact with target proteins, play a central role in identifying potential drug candidates. Furthermore, the use of pharmacophore modelling has become integral in optimizing lead compounds by mapping the essential features required for target binding. These computer-based strategies, by simulating molecular interactions, help predict the biological activity of compounds, offering valuable insights for lead optimization and the design of more effective therapeutics [27].

These advancements highlight the transformative role of computational tools in modern drug discovery, emphasizing their capability not only to accelerate the identification of promising candidates but also to deepen our understanding of complex biological systems. The integration of these methods into early-stage drug development represents a paradigm shift, paving the way for more efficient and precise therapies.

The application of virtual screening has led to the identification of several promising drug candidates across various therapeutic areas, including oncology, virology, and immunology. For instance, the repurposing of FDA-approved drugs through in silico methods was instrumental during the COVID-19 pandemic, revealing candidates that could inhibit critical viral proteins [28]. Similarly, hydroxychloroquine was repurposed as a potential anticancer agent for multiple myeloma through virtual screening, demonstrating its ability to target key pathways involved in the disease [29]. The work by Melge et al. [30] reports how disulfiram, an FDA-approved drug for alcoholism, was repurposed as a promising anticancer agent due to its ability to inhibit proteasomal activity and disrupt tumor cell metabolism. This discovery was supported by molecular docking studies, which demonstrated the strong binding affinity of disulfiram’s active metabolites to cancer-related targets such as proteasome subunits. Another example presented is ivermectin, originally approved as an antiparasitic agent, which exhibited potential as an inhibitor of nuclear transport proteins in cancer cells. Through virtual screening and binding affinity assessments, ivermectin was shown to interfere with key pathways involved in tumorigenesis [30]. These techniques are also widely applied to natural compounds, offering valuable insights into bioactive molecules derived from natural sources, as evidenced by the identification of eriocitrin and apigenin as potential inhibitors of carbonic anhydrase VA through virtual screening of Calabrian natural products [31].

Ritonavir is a Human Immunodeficiency Virus Type 1 (HIV-1) protease inhibitor, primarily used in the treatment of HIV infection. It has been repurposed in recent years for various diseases due to its ability to inhibit cytochrome P450 3A4 and P-glycoprotein, which play roles in drug metabolism and resistance. Beyond its use in HIV therapy, Ritonavir has shown considerable potential in oncology, where it has demonstrated antitumoral effects across multiple cancer types, including ovarian, prostate, lung, pancreatic, and breast cancers [32].

In this study, we aimed to identify selective bioactive ligands able to influence IL-20RA activity. Therefore, IL-20RA may represent a new therapeutic target for the management of TNBC unresponsive to current treatments [33,34]. Starting from 14 drugs screened using the ZINC15 [35] natural database and DrugBank database of FDA-approved compounds [36], we selected 4 promising candidates which were then evaluated for their biological activity, and of them, Ritonavir exerted anticancer activity against TNBC in vitro cell lines.

## 2. Results

### 2.1. Molecular Modelling Studies

Molecular dynamics simulation provided valuable insights into the structural stability and behavior of the IL-20/receptor complex. Throughout the 300 ns simulation, the system displayed remarkable stability, as evidenced by minimal energy fluctuations in the interaction diagram. The root mean square deviation (RMSD) values consistently remained within the range of 0.8 Å, indicating limited atomic positional changes and underscoring the integrity of the complex. This stability validated the reliability of the simulation for subsequent analyses (Figure 1).

Clustering of the trajectory yielded eight representative conformations, from which the most populated cluster, accounting for 240 frames, was identified as the most probable configuration based on its Boltzmann population of 99.457%. This representative structure, reflecting the dominant conformational state, was employed for downstream computational studies, including virtual screening and docking. The 1001 frames extracted from the molecular dynamics trajectory were superimposed on the selected cluster structure to identify the most representative frame of the cluster. This analysis revealed that frame 424, corresponding to a simulation time of 127.200 ns within the 300 ns trajectory, had the closest structural similarity to the cluster centroid, with a root mean square deviation (RMSD) of 1.66 Å.

### 2.2. Binding Sites Identification

Among the eight identified binding sites, three were selected for further investigation: two located at the interface between the IL-20 and the α-subunit, referred to as Binding Site-1 (BS-1), and one at the interface between IL-20 and the β-subunit, referred to as Binding Site-2 (BS-2), consistent with previous literature [16] (Figure 2)**.** In fact, in the study by Logsdon et al. [16], two critical binding sites, termed Site 1 and Site 2, were identified on the IL-20RA receptor. In particular, they described Site 1 as comprising two contact surfaces (Site 1a and Site 1b, which correspond to the two binding sites that we named BS-1). Site 1 is primarily responsible for the high-affinity interaction with IL-20, facilitating the initial cytokine–receptor engagement. Site 2 contributes to receptor dimerization and subsequent signal transduction. The structural integrity and functional relevance of these sites are crucial for IL-20-mediated signalling pathways [16].

Furthermore, Zeng et al. [37] have shown that IL-26, IL-10R2, and IL-20R1 share conserved structural motifs and functional domains between teleosts and higher vertebrates, including humans. Phylogenetic analyses demonstrate the clustering of these interleukins and their receptors with homologues in zebrafish and mammals, underscoring their ancient evolutionary lineage and functional significance [37].

Grids defining the 3D search space for virtual screening were generated using residues delineating the identified putative binding sites (BSs). Specifically, the residues for the α-subunit/IL-20 interface (referred to as BS-A) included Ile43, Phe45, Leu46, Thr81, Tyr109, Arg111, Ser126, Arg128, Phe129, Glu134, Arg227, and Arg228 for the α-subunit and Arg43, Ser47, Arg50, Gly51, Gln54, Ala55, Asp57, Gly58, Asp166, Ile167, and Gln170 for IL-20. For the β-subunit/IL-20 interface (referred to as BS-D), residues included Thr104, Thr106, Val107, Arg133, Gln134, Gln162, Lys210, and Ile212 for the β-subunit and Val34, Ile35, Ala36, Ala37, Asn38, Gln41, Ser111, Thr118, Ser115, Lys121, and Asp122 for IL-20. Additional grids were generated for the α-subunit using the same coordinates and dimensions as BS-A, but in one case considering only IL-20 (referred to as BS-B) and in the other considering only the α-subunit (referred to as BS-C). Similarly, two additional grids were generated for the β-subunit following the same approach, resulting in grids BS-E and BS-F (Figure 3). Thus, taken together, BS-A, BS-B, and BS-C are related to BS-1, while BS-D, BS-E, and BS-F are related to BS-2.

### 2.3. Virtual Screening

The virtual screening against the natural subset of the ZINC15 database was performed across the six grids defined for the α- and β-subunits, revealing a diverse range of potential ligands. For the α-subunit binding sites, the docking results identified 135 ligands for BS-A with docking scores ranging between −11.115 and −9.115 kcal/mol, 1044 ligands for BS-B with scores spanning −8.123 to −6.123 kcal/mol, and 569 ligands for BS-C with scores between −8.854 and −6.854 kcal/mol. Similarly, for the β-subunit, the docking process identified 389 ligands for BS-D with docking scores ranging from−10.367 to −8.367 kcal/mol, 150 ligands for BS-E with a docking score range from −9.620 to −7.620 kcal/mol, and 157 ligands for BS-F with a score range from −9.408 to −7.408 kcal/mol.

Subsequently, an analysis of shared ligands across different binding sites was conducted to identify molecules capable of simultaneously interacting with multiple regions of the protein complex. Several ligands exhibited significant versatility, binding to more than one site. Notably, three ligands—ZINC000256824195, ZINC000261498054, and ZINC000096085195—were found to interact with BS-A, BS-B, BS-C, and BS-D (Appendix A). Other ligands displayed similarly promising multi-site binding profiles, including ZINC000199624902 and ZINC000008551445, which interacted with BS-C, BS-F, BS-B, and BS-E (Appendix A), as well as ZINC000256824195, which bound to five grids, including BS-A, BS-D, BS-B, BS-C, and BS-F (Appendix A). Further ligands showed distinct combinations of interactions, such as ZINC000261495500 interacting with BS-A, BS-B, BS-C, and BS-E (Appendix A) and ZINC000096014977 binding to BS-A, BS-B, BS-C, and BS-F (Appendix A).

The DrugBank database was similarly screened for potential ligands targeting both subunits, leading to the identification of 15 ligands for binding site (BS)-A with docking scores ranging between −9.558 and −7.558 kcal/mol, 219 ligands for BS-D with scores between −8.297 and −6.297 kcal/mol, 8 ligands for BS-B ranging from −9.437 to −7.437 kcal/mol, 10 ligands for BS-E with a score range from −8.844 to −6.844 kcal/mol, 11 ligands for BS-C with score from −8.950 to −6.950 kcal/mol, and 7 ligands for BS-F with a score spanning from −9.297 to −7.297 kcal/mol. Shared ligand analysis revealed multiple compounds capable of interacting with multiple binding sites. Lypressin was found to bind to BS-A, BS-B, and BS-C (Appendix A), while Ritonavir (Table 1) and Fenoterol (Appendix A) simultaneously interacted with BS-A and BS-D. Other notable ligands included Goserelin, which interacted with BS-B and BS-F (Appendix A); Desmopressin, which engaged with BS-C, BS-F, BS-B, and BS-E (Appendix A); and Atosiban (Appendix A) and Triptorelin (Appendix A), which bound to both BS-C and BS-E.

The identified molecules were evaluated for their feasibility in further experimental validation. This involved a careful analysis based on their chemical structures to exclude compounds that were easily hydrolyzable or not commercially available. From the remaining candidates, binding modes were thoroughly examined to prioritize molecules that exhibited the most promising interactions with IL-20R type 1. Additionally, the selection of compounds was guided by a thorough review of the literature, which highlighted their mechanisms of action and their therapy indications, as shown in Table 1 and in Appendix A. Based on this selection process, four molecules were identified: Goserelin, Triptorelin, Ritonavir, and Fenoterol. Notably, Goserelin [36,40,41] and Triptorelin [36,42,43] were selected due to their established approval as anticancer agents, underscoring their clinical relevance. Ritonavir [32,36,38] was included based on extensive documentation in the literature highlighting its in vitro antitumor activity. Fenoterol [36,44], in contrast, stood out due to its significantly lower molecular weight compared to the other compounds, a distinct structural feature that suggests a potentially unique mechanism of action or interaction, warranting further investigation.

### 2.4. Visual Inspection and Ligand Interactions

The four compounds were subsequently analyzed through visual inspection to assess their characteristics. Goserelin demonstrated good interactions with both BS-B and BS-F. In particular, IL-20RA interacts with Goserelin through different hydrogen bonds established with Arg128, Gln107, Tyr130, and Glu134, and among them, the latter forms an additional salt bridge. From the BS-F binding site, the complex is stabilized through several hydrogen bonds established with Val34, Lys121, Thr118, Ile119, Ala 36, and Glu41, which also interact through a salt bridge (Appendix A). Triptorelin showed good affinities for BS-C and BS-E. It is able to interact with BS-C through several hydrogen bonds established with Ser47, Arg50, Gly51, Gln54, Asp57, Glu164, and Asp166; two different Pi–cation interactions with Arg50 and Arg43; and a salt bridge with Asp57. On the other hand, the compound forms favorable interactions with IL-20RB, such as hydrogen bonds with Asp101, Asp102, Tyr109, Pro226, Gln162, Glu164, Tyr215, and Arg133 and two additional salt bridges mediated by Asp101 and Asp102 (Appendix A). Finally, both Fenoterol and Ritonavir bind at the interface of both the IL-20/IL-20RA complex (BS-A) and the IL-20RB/IL-20 complex (BS-D), engaging with residues from the IL-20RA and IL-20RB subunits and IL-20. Fenoterol is stabilized in BS-A through four hydrogen bonds with Arg43, Arg128, Asp166, and Glu134, which also form a salt bridge; moreover, Phe129 is involved in a Pi-Pi stacking. In BS-D, Fenoterol interacts with Asp122, Glu41, and Gln134 through three hydrogen bonds, while an additional Pi–cation interaction is mediated by Lys12 (Appendix A).

In BS-A, Ritonavir establishes four distinct hydrogen bonds with Arg128 and Glu134 of IL-20RA, while Arg50 of IL-20 participates in an additional hydrogen bond with the compound. On the BS-D side, a hydrogen bond is formed with Gln134 of the IL-20RB subunit. A greater number of interactions are observed with IL-20, particularly involving Ala36, Asn38, Lys121, and Asp122, which form hydrogen bonds. Notably, Lys121 also engages in a Pi–cation interaction (Figure 4).

### 2.5. Ritonavir Exerts Anti-Growth Activity Against TNBC Cells

To confirm the above-reported in silico findings, we first analyzed the effects of various concentrations of the four selected ligands on the proliferation of two human TNBC distinct cells, MDA-MB-157 and MDA-MB-231. As visualized in the time–dose–response curves of the four selected drugs, only Ritonavir displayed significant antiproliferative effects (Figure 5 and Appendix A).

In particular, for Ritonavir, at the concentration of 50 µM, after 24 h of treatment, a significant reduction in cell viability of about 40% vs. untreated cells was determined in both cell lines, reaching ~60% and 80% over the control after 48 and 72 h, respectively (Figure 5a). In terms of variance, for the other ligands, the antiproliferative effect did not appear to be linear or dose- and time-dependent (Appendix A).

Since Ritonavir exerted the best antiproliferative effects, additional analyses were only performed on this compound. To determine the patterns of cell cycle progression in Ritonavir-mediated cellular effects, we performed cytofluorimetric analysis. The results showed a significant slowing of the cell cycle S phase and an increase in cell population in the sub-G0 phase in cells exposed to Ritonavir 50 µM for 24 h (Figure 5b,c). Moreover, this IC_50_ value corresponds to the one previously reported in other BC studies, in which the tumor-inhibitory dose of Ritonavir was well tolerated with acceptable toxicity in xenografts mice [45,46].

### 2.6. IL20RA Is Involved in the Antiproliferative Effect Induced by Ritonavir in TNBC Cells

The biological assays conducted in this study specifically targeted the IL-20RA subunit due to its unique role in interacting with IL-20. IL-20RA, when heterodimerized with IL-20RB, forms the IL-20R type 1 complex, which is essential for IL-20 signalling. In contrast, IL-20RB has a broader interaction spectrum, as it can pair with IL-20RA to form the IL-20R type 1 complex or with IL-22R to create the IL-20R type 2 complex. Both complexes are capable of binding IL-20 [14]. This receptor specificity underscores why IL-20RA is a more suitable target for interventions aimed at inhibiting IL-20R type 1 formation. Targeting IL-20RA provides specificity for disrupting IL-20-mediated signalling without affecting IL-20RB’s participation in the IL-20R type 2 complex or other cytokine interactions.

To understand whether the activation of IL-20RA was responsible for Ritonavir’s cytostatic effect, TNBC cells were preincubated for 2 h with IL-20 followed or not by treatment with Ritonavir 50 µM for 24 h. As shown in Figure 6, pretreatment with IL-20 (200 ng/mL) significantly counteracted the growth-inhibiting action of Ritonavir (~20% and 30% over Ritonavir alone in MDA-MB-157 and MDA-MB-231 cells, respectively; *p* < 0.05, *p* < 0.01).

In addition, we transfected MDA-MB-157 and MDA-MB-231 cells with three different and specific hIL20RA siRNAs (sihIL20RA a, b, c). As shown in Figure 7a, the strongest decrease in the levels of IL20RA protein expression was obtained with sihIL20RA c. After transfection with sihIL20RA c or Stealth RNAi Negative Control Duplexes (NC siRNA), TNBC cells were treated with Ritonavir 10 and 50 µM for 24 h. The growth-inhibiting action of Ritonavir in cells transfected with NC siRNA was significantly restored by cell transfection with sihIL20RA c (~20% over NC siRNA treated with Ritonavir 50 µM in both cell lines; °° *p* < 0.01) (Figure 7b).

### 2.7. Thermodynamic Analysis

The biological data were further supported by the Molecular Mechanics/Generalized Born Surface Area (MM-GBSA) method. This analysis, specifically conducted on the BS-A binding site, provided deeper insights into the compound’s inhibitory potential. The resulting value of −74.20 kcal/mol highlighted the thermodynamic stability of the IL-20RA/Ritonavir/IL-20 complex.

## 3. Discussion

TNBC represents an aggressive BC subtype associated with a generally poor BC outcome and high rate of recurrence and usually managed by conventional approaches, such as surgery and chemo- and radiotherapy, but lacking effective targeted therapies. Hence, finding new therapeutic strategies is fundamental for TNBC management.

In the search for new and alternative approaches, encouraging results derive from the application of compounds able to target the TME, a heterogeneous milieu that influences tumor behavior and therapeutic outcomes [12,47]. In this context, both immune and cancer cells create an environment that inhibits immunosurveillance and promotes tumor progression. Among the secreted factors, ILs are key players in the cytokine network in TME, and recently, it has been reported that deregulation of IL-20 subfamily member signalling pathways can impact tumor growth and progression as well as metastatic spread in several types of cancer cells. Upon binding to their heterodimeric receptors, all IL-20 subfamily cytokines activate the JAK–STAT pathway, and mainly STAT3 by inducing proliferation, survival, epithelial–mesenchymal transition (EMT), and migration of transformed cells [17]. Noteworthy, extensive research reported the ability of general anesthetics to influence the secretion of inflammatory cytokines such as IL-6, IL-10, and TNF-α in the TME of BC. This, in turn, could heighten the risk of the development of lung cancer metastases after surgery. For example, in BC cases involving lung metastases, sevoflurane exposure during surgery has been observed to influence the lung TME through IL6/JAK/STAT3 signalling pathway regulation. Additional in vivo and in vitro studies are needed to clarify the mechanisms by which anesthetics affect cancer cells and the immune system in the TME [48,49].

Several works demonstrated that both IL-20 and IL-20RA expression levels are elevated in different types of preclinical models of cancer cells including BC. The expression of IL-20 in BC tissue is not only associated with a higher mitotic rate but also correlated with advanced tumor stages and bone metastasis. The application of a monoclonal antibody against IL-20 (anti-IL-20 mAb) in a mouse BC model provided an unfavorable microenvironment for tumor cells to colonize and grow [24,25,26,50]. The aforementioned investigations reveal the biological significance of IL-20RA in tumor cells, suggesting its potential use not only as a biological marker in diagnostic analysis but also as a therapeutic target.

In our study, the putative binding sites of the IL-20R type 1 receptor were mapped, and the amino acid residues defining these sites were subsequently used to construct grids for virtual screening. This virtual screening identified fourteen ligands capable of simultaneously interacting with both IL-20RA and IL-20RB, being well accommodated into the two defined binding sites.

Supporting in silico data, the antiproliferative effect of the top four selected compounds (Fenoterol, Goserelin, Triptorelin, and Ritonavir) in in vitro models of TNBC was investigated. Only Ritonavir was able to reduce the proliferation of MDA-MB-157 and MDA-MB-231 cancer cells at a micromolar concentration, which corresponds to the same dose used in both breast and other tumor cellular models [45,51]. This effect was associated with a decrease in cell cycle S phase and sub-G0 arrest, with data suggesting a cytostatic effect of the drug. Fenoterol, Goserelin, and Triptorelin showed no remarkable influence on cell proliferation, leading to their exclusion in further investigations.

Ritonavir’s antitumoral properties stem from several mechanisms. In vitro studies have shown that it can inhibit cancer cell proliferation and induce apoptosis through various pathways and mechanisms including cell cycle modulation and induction of endoplasmic reticulum (ER) stress [51,52,53,54]. Ritonavir has also been shown to affect mitochondrial function, leading to reduced ATP production and influencing key signalling pathways involved in cancer progression, such as the Akt and Hsp90 pathways [45,52,55]. Additionally, Ritonavir interferes with proteasome function, particularly by accumulating tumor suppressor proteins like p21WAF1 and modulating the degradation of key cell cycle regulators, thus contributing to apoptosis in tumor cells [38]. Furthermore, it sensitizes resistant cancer cells to traditional chemotherapeutic agents such as gemcitabine and docetaxel, enhancing the efficacy of these drugs [32]. Ritonavir’s potential for cancer treatment is further supported by its ability to inhibit P-glycoprotein expression and activity, a major mediator of multidrug resistance (MDR) in cancer cells. In addition, it inhibits BC-resistant protein (BCPR) efflux, enhancing intracellular drug concentration [56,57]. These actions improve the accumulation of chemotherapeutic agents within tumor cells, making Ritonavir a valuable candidate for overcoming drug resistance [54,55]. The aforementioned antitumoral effects highlight Ritonavir’s potential as a repurposed drug for cancer therapy.

The interplay between Ritonavir and TME has not previously been investigated. In this study, we proposed a molecular target involved in the growth arrest of TNBC cells in response to Ritonavir.

Given the specificity of IL-20RA in interacting with IL-20-mediated signalling, we decided to verify Ritonavir’s capability to interfere with IL-20RA and its ligand IL-20. We tested its effect on cellular proliferation after pretreatment with IL-20, observing that IL-20 significantly counteracted the ability of Ritonavir to inhibit cellular proliferation. This result is consistent with a report regarding the role of IL-20 in promoting the growth of BC cells [20]. In addition, we demonstrated both that IL-20RA is expressed in TNBC cells and its knockdown substantially restored proliferation in Ritonavir-treated TNBC cells. The Molecular Mechanics/Generalized Born Surface Area (MM-GBSA) method supported the biological data, demonstrating favorable stability of the IL-20RA/Ritonavir/IL-20 complex.

To date, this study represents the first report establishing a correlation between Ritonavir’s ability to inhibit the proliferation of TNBC cells and its interaction with IL-20RA.

In conclusion, we showed that Ritonavir affects the proliferation of TNBC cells, influencing IL-20RA activity. These data open new insight into the potential repurposing of Ritonavir in the treatment of cancer and suggest the investigation of this molecule in the management of TNBC. Further studies in in vivo models will be needed in order to study pharmacological aspects strictly related to Ritonavir’s anticancer applications, including the management of TNBC and other tumors characterized by high levels of IL-20RA expression.

## 4. Materials and Methods

### 4.1. Structural Model Selection and Optimization

Starting from the three-dimensional structure of the human IL-20RA/IL-20/IL-20RB ternary complex deposited in the Protein Data Bank (PDB) [58,59] with the PDB code 4DOH [16], in silico studies were carried out. The protein structure was prepared using the Protein Preparation Wizard tool in Maestro (Schrödinger, LLC: New York, NY, USA) [60]. Crystallographic buffer components were removed, and a single chain was retained for each subunit. Hydrogen atoms were added, missing side chains were reconstructed using the Prime module, and the protonation states of side chains were assigned at pH 7.4. Energy minimization was performed to optimize the prepared structure, employing the OPLS_2005 force field.

### 4.2. Molecular Dynamics Simulations

The optimized complex was subjected to 300 ns of molecular dynamics (MD) simulations using Desmond (Schrödinger, LLC: New York, NY, USA) [61]. The protein complex was solvated in an orthorhombic box with a 10 Å buffer of TIP4P (Transferable Intermolecular Potential 4-Point) water molecules, and Cl- ions were added to neutralize the system’s net charge. The simulations were carried out under an isothermal–isobaric (NPT) ensemble, maintaining a temperature of 300 K and a pressure of 1 bar using the Berendsen thermostat and barostat.

### 4.3. Trajectory Clustering and Energy Minimization

The resulting trajectory was clustered based on RMSD, yielding eight representative structures. Each representative structure underwent energy minimization through MacroModel (Schrödinger, LLC: New York, NY, USA) [62] with the OPLS_2005 force field, performing 10,000 iterations for each structure. These minimized structures were evaluated according to the Boltzmann population (Schrödinger, LLC: New York, NY, USA) [63], calculating the percentage probability of existence of each cluster. Subsequently, the selected cluster was superimposed and compared, in terms of RMSD, with each of the 1000 frames obtained from the molecular dynamics trajectory, in order to identify a frame corresponding to the structure and determine approximately the simulation time with which it is associated.

### 4.4. SiteMap and Grid Generation

The selected structure was used for binding site mapping with SiteMap (Schrödinger, LLC: New York, NY, USA) [64]. SiteMap and Glide Grid generation (Schrödinger, LLC: New York, NY, USA) [65] provided quantitative and graphical insights to identify potential binding pockets and to generate grids used for virtual screenings.

### 4.5. Virtual Screening Protocols

Virtual screening was performed using two different databases: ZINC15 [35], containing 307,814 natural compounds, and DrugBank [36], comprising 2478 FDA-approved drugs. Ligands were prepared with LigPrep (Schrödinger, LLC: New York, NY, USA) [39], hydrogens were added, salts were removed, ionization states were calculated at pH 7.4 using Epik, and energy minimization was performed using the OPLS_2005 force field. Molecular docking was conducted for each of the six generated grids using the Glide Standard Precision (SP) (Schrödinger, LLC: New York, NY, USA) protocol [65,66,67]. Ten docking poses were generated for each ligand, and theoretical binding affinities were reported as docking scores (kcal/mol). Compounds from each BS were evaluated based on their docking scores, with a cutoff set at 2 kcal/mol below the best pose.

### 4.6. Thermodynamic Analysis: MM-GBSA

The IL-20RA/Ritonavir/IL-20 complex was estimated by means of the Molecular Mechanics Generalized Born/Surface Area (MM-GBSA) method (Schrödinger, LLC: New York, NY, USA) [63]. Specifically, it uses the calculated molecular mechanics energies and the variable-dielectric generalized Born model (VSGB) as continuum solvation models to compute the difference between the energy of the bound complex and the energies of the unbound protein (ΔG_Bind_).

### 4.7. Cell Culture, Proliferation and Cell Cycle Assays

MDA-MB-157 and MDA-MB-231, in vitro models of human TNBC widely used for their high ability for growth and progression, were purchased from the American Type Culture Collection (Manassas, VI, USA), cultured in DMEM, and maintained at 37 °C in humidified 5% CO_2_ as previously described [68]. Cell proliferation was evaluated by the 3-(4,5-dimethylthiazol-2-yl)-2,5-diphenyltetrazolium bromide (MTT) assay. MDA-MB157 and MDA-MB231 were seeded in 96-well plates at a density of 3.5 × 10^3^ or 4 × 10^3^, respectively. Untreated cells and those pretreated with IL-20 (PEPROTECH, Thermo Fisher Scientific Inc., Waltham, MA, USA) were incubated with Goserelin, Triptorelin, Fenoterol, and Ritonavir (Molport EU, Riga, Latvia), diluted in DMSO as the vehicle, for 24, 48, and 72 h. The solubilized products were quantified with a microplate spectrophotometer (Varioskan Lux, Thermo Fisher Scientific Inc.) at a wavelength of 540 nm and a reference wavelength of 690 nm. Results are expressed as percentages over untreated cells (vehicle), indicated as control.

By flow cytometry, we assessed cell cycle distribution. After 24 h of treatment with ritonavir 10 and 50 µM, cells were harvested, fixed in 70% ethanol, and stored at −20 °C. Then, the cells were washed twice with PBS and incubated with RNase and DNA intercalating dye propidium iodide (Sigma Aldrich, Milan, Italy). Cell cycle phase analysis was performed using the flow cytometer ACCURITM C6 (Becton Dickinson, San Jose, CA, USA).

### 4.8. hIL-20RA Silencing

Three different small interfering RNA (siRNA) oligonucleotides targeting IL-20RA (a. 5′-GCCCGCAAACGUUACAGUACUCAUA-3′; b. 5′-GACCUUCCUGUUUCCAUGCAACAAA-3′; c. 5′-CCUCAAUAUCUCGGAUGAUUCUAAA-3′) were transiently transfected into cells using lipofectamine RNAiMAX (Thermo Fisher Scientific Inc.), following the manufacturer’s instructions.

Cells were plated with medium containing FBS 10% in 6-well plates (MDA-MB-157, 150 × 10^3^/well and MDA-MB-231 180 × 10^3^/well) and reached 60–80% confluence at the time of transfection. After 72h, transfection efficiency was evaluated by the Western blot assay. In all experiments, cells transfected with Stealth RNAi Negative Control Duplexes (Thermo Fisher Scientific Inc.) were used as control (indicated as NC siRNA).

After hIL-20RA silencing, cells were exposed to Ritonavir 10 and 50 µM for 24 h and cell growth was evaluated by the MTT assay. Results are expressed as percentages over control (NC siRNA) untreated or treated with Ritonavir.

### 4.9. Western Blot Analysis

Cell monolayers were solubilized in a pre-heated 1% SDS solution as previously described [69] and proteins were determined with the BCA reagent (Pierce). An amount of 30 µg of proteins was run on a 9% SDS-PAGE gel; transferred to PVDF membranes (VWR, Milan, Italy); blocked with phosphate-buffered saline, 0.1% Triton 10%, and 5% non-fat dry milk (T-PBS); and incubated overnight with affinity-purified anti-IL-20RA (Thermo Fisher Scientific Inc.) and anti-GAPDH antibodies (Merk Life Sciences, Milan, Italy), diluted 1:1000 and 1:10,000, respectively. The membranes were incubated with horseradish peroxidase-conjugated secondary antibodies (BD Biosciences, Milan, Italy) and signals were detected by using a chemiluminescence system (ECL Plus, Revvity, Milan, Italy) to visualize proteins.

### 4.10. Statistical Analysis

Data from cell proliferation experiments were analyzed by one-way ANOVA followed by the Tukey–Kramer multiple comparisons test. All results are expressed as mean ± standard deviation (SD) and were considered statistically significant at *p*-values lower than 0.05. All statistical analyses were performed using GraphPad Prism version 9.3.0 statistical software (GraphPad Software Inc., San Diego, CA, USA).

## Figures and Tables

**Figure 1 ijms-26-01285-f001:**
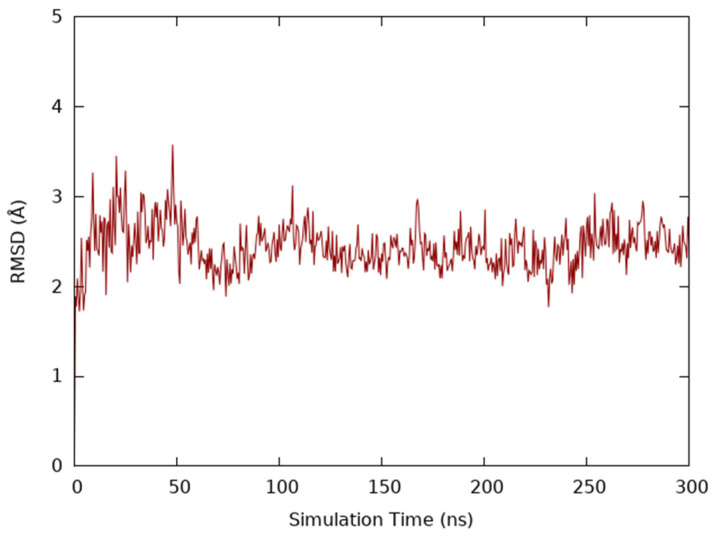
RMSD (Å) trend calculated on the IL-20R type 1 backbone over 300 ns of simulation time.

**Figure 2 ijms-26-01285-f002:**
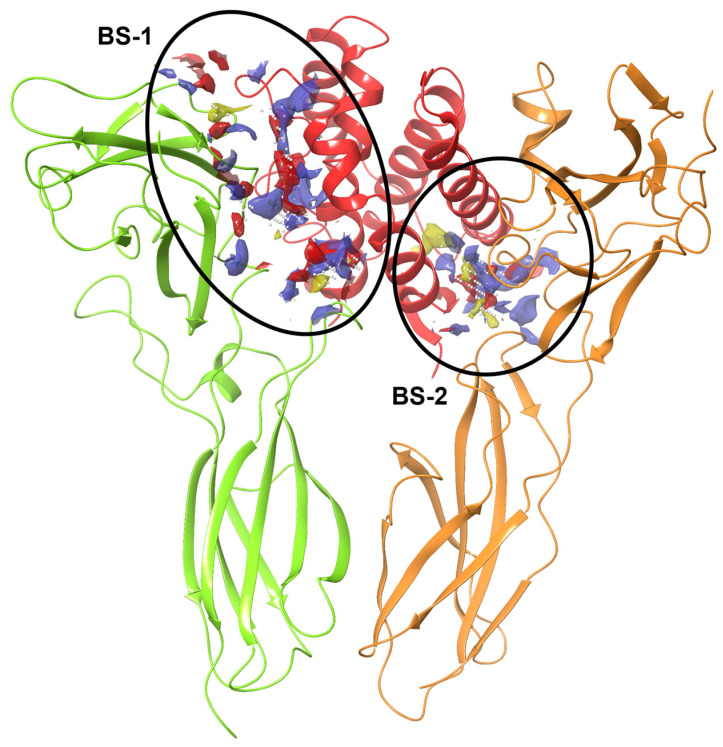
Binding sites of IL-20R type 1. Pockets, referred to as BS-1 and BS-2, are shown with predicted hydrophobic regions in yellow mesh, hydrogen bond donor surfaces in blue mesh, and acceptor surfaces in red mesh. IL-20RA is represented as green ribbon, IL-20RB as orange ribbon, and IL-20 as red ribbon.

**Figure 3 ijms-26-01285-f003:**
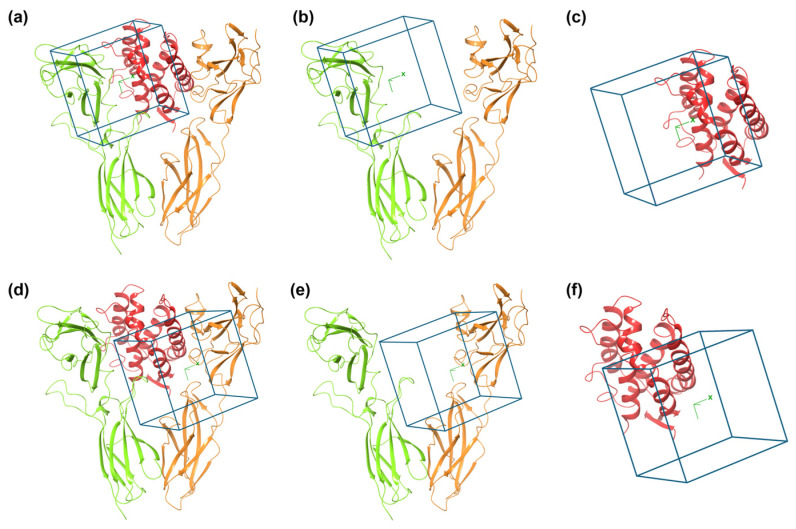
Representation of the six grids defining the putative binding sites (BFs). (**a**): BS-A. (**b**): BS-B. (**c**): BS-C. (**d**): BS-D. (**e**): BS-E. (**f**): BS-F. IL-20RA is represented as green ribbon, IL-20RB as orange ribbon, and IL-20 as red ribbon.

**Figure 4 ijms-26-01285-f004:**
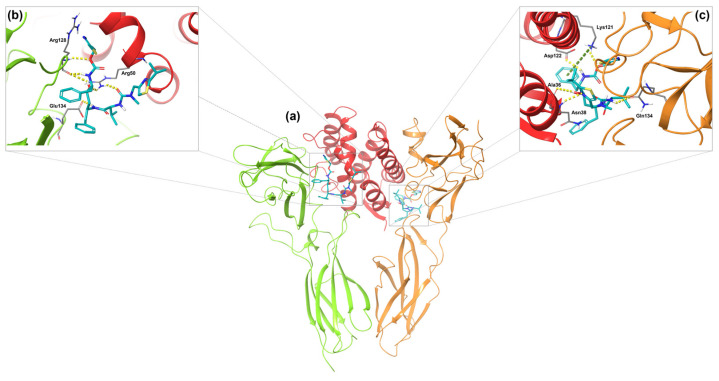
(**a**): Three-dimensional representation of Ritonavir (shown as cyan sticks) bound to the binding sites of IL-20R type 1. (**b**,**c**): These panels provide detailed views of the binding modes and ligand interactions in BS-A and BS-D, respectively. IL-20RA is depicted as green ribbon, IL-20RB as orange ribbon, and IL-20 as red ribbon. Amino acid residues interacting with the compound are represented as grey carbon sticks. π–cation and hydrogen bonding interactions are shown as green and yellow dashed lines, respectively.

**Figure 5 ijms-26-01285-f005:**
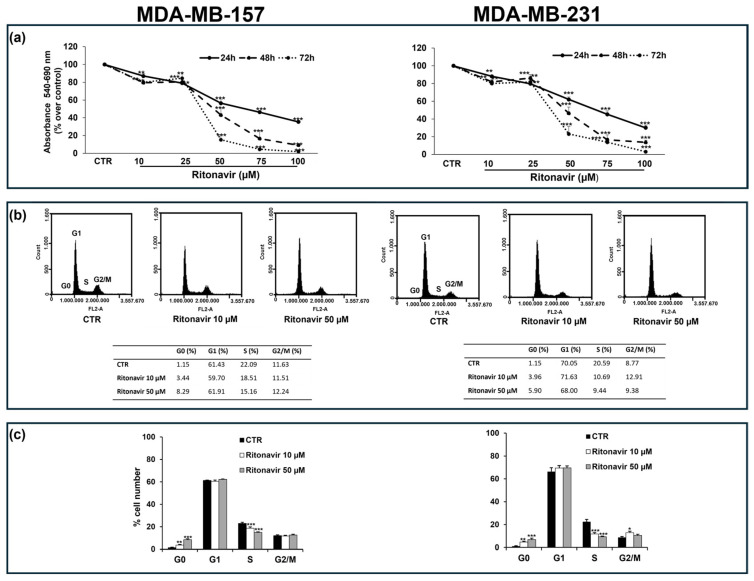
Effects of Ritonavir on cell viability and cell cycle of MDA-MB-157 and MDA-MB-231 cells. (**a**): Cell viability evaluated by MTT after 24, 48, and 72 h of incubation with different concentrations of Ritonavir. Results are presented as mean ± SD of three independent experiments performed eight times (see also Appendix A). (**b**): Plots are representative of an independent biological replicate, n = 3, and show the cell cycle distribution obtained by cytofluorimetric analysis after treatment with the indicated concentrations of Ritonavir for 24 h. The percentage of cells in each population is shown in the tables below the plots. (**c**): Data are presented as mean ± SD of three independent experiments. Statistical analysis was performed using the Tukey–Kramer multiple comparisons test. * *p*< 0,05, ** *p* < 0.01, *** *p* < 0.001 vs. control. Control, indicated as CTR, represents the untreated cells.

**Figure 6 ijms-26-01285-f006:**
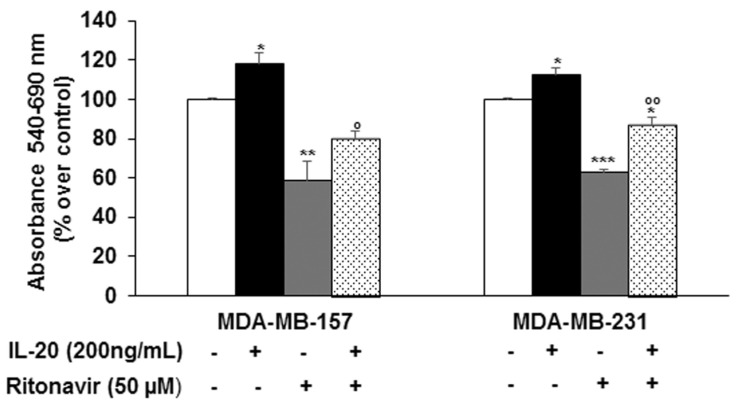
Effect of IL-20, Ritonavir, and their combination on cell viability of MDA-MB-157 and MDA-MB-231 cells. Cells were preincubated with IL-20 (200 ng/mL) for 2 h followed by the addition or not of Ritonavir (50 µM) and incubation for an additional 24 h. Cell viability was evaluated by MTT and the results are mean ± SD of three independent experiments performed eight times. Statistical analysis was performed using the Tukey–Kramer multiple comparisons test. * *p* < 0.05, ** *p* < 0.01, *** *p* < 0.001 vs. control; ° *p* < 0.05, °° *p* < 0.01 vs. Ritonavir.

**Figure 7 ijms-26-01285-f007:**
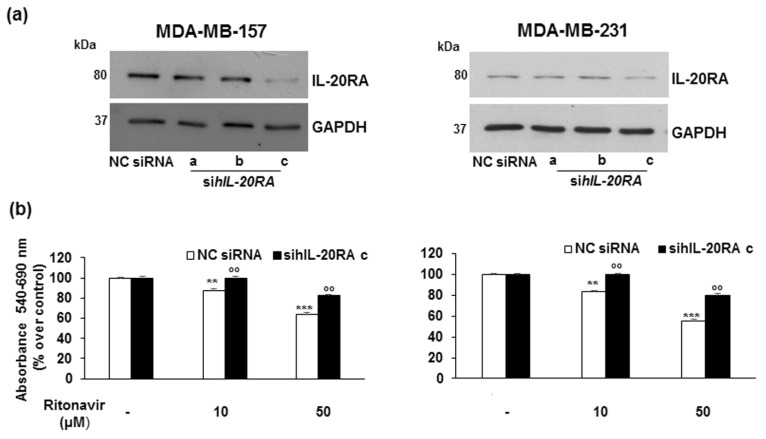
The knockdown of IL-20RA restored proliferation in Ritonavir-treated TNBC cells. (**a**): Silencing of hIL-20RA by three siRNAs in MDA-MB-157 and MDA-MB2-31 cells. Western blot analysis is representative of three different experiments, as described in Materials and Methods. GAPDH was used as an internal control. (**b**): Cells, treated with sihIL-20RA c or siRNA sequence control (NC siRNA), were exposed for 24 h to Ritonavir 10 and 50 µM. Results are mean ± SD of three independent experiments performed eight times. Statistical analysis was performed using the Tukey–Kramer multiple comparisons test. ** *p* < 0.01, *** *p* < 0.001 vs. NC siRNA; °° *p* < 0.01 vs. NC siRNA treated with Ritonavir 10 or 50 µM.

**Table 1 ijms-26-01285-t001:** Two-dimensional structure, docking score (kcal/mol), molecular weight (Da), logP, molecular target, and therapeutic application of Ritonavir [32,36,38] within binding sites BS-A and BS-D. The chemical structure was created by Maestro Graphical Interface [39].

	Docking Scores (kcal/mol)					
	BS-A	BS-D	MW	logP	Molecular Target	Therapeutic Application
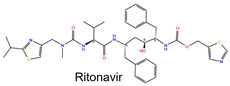	−7.72	−6.66	720.94	4.24	HIV protease	HIV

## Data Availability

The original contributions presented in this study are included in the article/Appendix A. Further inquiries can be directed to the corresponding author.

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
