# Peer review of "Rational Identification of Ritonavir as IL-20 Receptor A Ligand Endowed with Antiproliferative Properties in Breast Cancer Cells"

_ijms, 2025, doi:10.3390/ijms26031285_

Round 1
Reviewer 1 Report
Comments and Suggestions for Authors
The manuscript titled "Rational Identification of Ritonavir as IL-20 Receptor A Ligand Endowed with Antiproliferative Properties" explores the novel application of Ritonavir, an HIV-1 protease inhibitor, as a potential therapeutic agent targeting the IL-20RA in triple-negative breast cancer (TNBC). Through a combination of in silico methods and experimental validation, the study uncovers significant insights into Ritonavir’s mechanisms of action and its ability to inhibit TNBC cell proliferation. The manuscript is well-organized and presents robust data; however, it requires additional elaboration in the Introduction, Results, and Discussion sections to contextualize findings and enhance clarity.
Detailed Comments
Abstract
Line 6: Clarify how IL-20RA contributes to tumor progression and its relevance as a drug target.
Line 10: Include specific outcomes, such as the percentage reduction in TNBC cell proliferation achieved by Ritonavir.
Introduction
3 line 20: Provide updated statistics on cancer in general as well as this cancer type prevalence, including survival rates, to highlight the critical need for prognostic biomarkers. Cite “Cancer statistics, 2024, 2024”. Then give intro in cancer therapy in general, cite NIH paper “Cancer treatments: Past, present, and future, 2024” (PMID: 38909530)for more information.
Line 35: Expand on the molecular mechanisms through which IL-20RA signaling drives tumor progression, including its role in the JAK–STAT pathway.
4. Line 55: Provide examples of how in silico drug repurposing has successfully identified anticancer agents in other malignancies. Discuss also the microenvironment of cancer cells, how they can affect immune cells, recent studies in breast cancer microenvironment should be mentioned, such as “Identification of the novel exhausted T cell CD8 + markers in breast cancer, 2024”
5. Line 70: Highlight gaps in current TNBC therapies that the study aims to address, emphasizing the lack of effective targeted treatments.
Materials and Methods
6. Line 100: Include details on the software and parameters used for virtual screening, particularly for docking score calculations. Cite previous studies using docking to support the approach, such as “Identification of molecular targets of Hypericum perforatum in blood for major depressive disorder: a machine-learning pharmacological study, 2024,Chebulinic Acid isolated from Aqueous Extracts of Terminalia chebula Retz inhibits Helicobacter pylori infection by potential binding to cag A protein and regulating adhesion, 2024,Isoform-specific N-linked glycosylation of NaV channel α-subunits alters β-subunit binding sites, 2025”
7. Line 130: Explain how potential ligands were selected for experimental validation, focusing on their predicted binding affinities and structural properties.
8. Line 150: Specify the rationale behind the selection of MDA-MB-157 and MDA-MB-231 cell lines for in vitro experiments. Mention some previous studies using MDA-MB-231 for cell cycle analysis might help, such as “The Role of Transient Receptor Potential Melastatin 7 (TRPM7) in Cell Viability: A Potential Target to Suppress Breast Cancer Cell Cycle, 2020,Effects of local anesthetics on breast cancer cell viability and migration, 2018”
Results
9. Line 200: Discuss the biological relevance of the identified binding sites on IL-20RA, including their conservation across species.
10. Line 240: Provide additional statistical analysis or graphical representation of docking scores for the top-ranked ligands.
11. Line 280: Elaborate on the observed differences in antiproliferative effects between Ritonavir and the other tested ligands.
12. Line 320: Highlight potential off-target effects of Ritonavir and how these were mitigated in the study design.
Discussion
13. Line 400: Discuss the potential for Ritonavir to overcome drug resistance in TNBC, referencing its known effects on P-glycoprotein inhibition. Suggest potential future study for drug resistance target discovery, such as CRISPR screening, and how these can help, suggest to refer to “CRISPR screening and cell line IC50 data reveal novel key genes for trametinib resistance, 2025”
14. Line 450: Address the limitations of using 2D in vitro models and propose follow-up studies involving 3D cultures or in vivo validation. Suggest future studies that could validate these findings in patient-derived xenograft models or larger cohorts. Previous studies using xenograft models of cancer should be mentioned, such as “Comparing volatile and intravenous anesthetics in a mouse model of breast cancer metastasis, 2018”
15. Line 480: Explore the broader applicability of IL-20RA inhibitors in other cancers characterized by high IL-20RA expression.
Add additional sicussion: Studies suggested that anesthetics during surgery treatment can impact cancer, reported in a series of work by Prof Lin’s group: “Effects of local anesthetics on breast cancer cell viability and migration, 2018,Effects of local anesthetics on cancer cells, 2020,Effect of Propofol on breast Cancer cell, the immune system, and patient outcome, 2018,Lidocaine Suppresses Viability and Migration of Human Breast Cancer Cells: TRPM7 as A Target for Some Breast Cancer Cell Lines, 2021,The Potential Effect of General Anesthetics in Cancer Surgery: Meta-Analysis of Postoperative Metastasis and Inflammatory Cytokines, 2023,Potential Therapeutic Application of Local Anesthetics in Cancer Treatment, 2022” These should be emphasized and discuss if the anesthetics impact involved in the mechanisms discussed in this study
Figures and Tables
16. Figure 2: Include annotations or labels on binding site visualizations to improve reader comprehension.
17. Figure 5: Present raw data for cell viability assays in a supplementary table to ensure reproducibility.
18. Table 1: Add additional information about the chemical structures and properties of the four ligands tested.
Author Response
Review Report Form #1
Comments and Suggestions for Authors
The manuscript titled "Rational Identification of Ritonavir as IL-20 Receptor A Ligand Endowed with Antiproliferative Properties" explores the novel application of Ritonavir, an HIV-1 protease inhibitor, as a potential therapeutic agent targeting the IL-20RA in triple-negative breast cancer (TNBC). Through a combination of in silico methods and experimental validation, the study uncovers significant insights into Ritonavir’s mechanisms of action and its ability to inhibit TNBC cell proliferation. The manuscript is well-organized and presents robust data; however, it requires additional elaboration in the Introduction, Results, and Discussion sections to contextualize findings and enhance clarity.
Detailed Comments
Abstract
- Line 6: Clarify how IL-20RA contributes to tumor progression and its relevance as a drug target.
ANSWER: We thank the referee for his/her comment. Following the reviewer’s suggestion, we have included more information about how IL-20RA contributes to tumor progression and its relevance as a drug target, by adding the following sentence: “Recent studies have highlighted the involvement of IL-20 receptor subunit alpha (IL-20RA) signalling in several cancers, including BC in which IL-20RA is highly ex-pressed associating with poor prognosis and influencing tumoural characteristics such as proliferation, cell death, invasiveness and TME activity. Therefore, elucidating the role of the IL-20RA signalling pathway could form the basis for developing new therapeutic strategies. This study aimed to identify selective bioactive ligands able to affect IL-20RA activity.” (Section Abstract, lines 22-25).
- Line 10: Include specific outcomes, such as the percentage reduction in TNBC cell proliferation achieved by Ritonavir.
ANSWER: Thanks for the suggestion. The following sentence has been reported in the revised version of the manuscript: “about 40% at 50 µM, p<0.001”. (Section Abstract, lines 31-32).
Introduction
- 3 line 20: Provide updated statistics on cancer in general as well as this cancer type prevalence, including survival rates, to highlight the critical need for prognostic biomarkers. Cite “Cancer statistics, 2024, 2024”. Then give intro in cancer therapy in general, cite NIH paper “Cancer treatments: Past, present, and future, 2024” (PMID: 38909530) for more information.
ANSWER: Thanks for the suggestion. In the revised version of the manuscript, we added this information in Section Introduction, lines 39-42, i.e. “Cancer is a major societal, public health, and economic problem in the 21st century that caused 20 million new cases in the year 2022 alongside 9.7 million deaths [1]. Breast cancer (BC) is the most frequently diagnosed cancer in women globally and the leading cause of cancer death worldwide, in 157 countries for incidence and 112 for mortality [1,2]”. The suggested reference #1, which was already present in the first version of the manuscript.
- Line 35: Expand on the molecular mechanisms through which IL-20RA signaling drives tumor progression, including its role in the JAK–STAT pathway.
ANSWER: Thanks for the suggestion. In the revised version of the manuscript, we have provided more information on the molecular mechanisms through which IL-20RA signaling drives tumor progression and its role in the JAK-STAT pathway, by including the following sentences: “Recent data have associated IL-20RA signalling with hallmarks of cancer, including regulation of proliferative signalling, resistance to cell death, activation of cellular mi-gration and invasion [24–26]. Upon ligand binding, IL-20R type 1 phosphorylates ac-tivating JAK that in turn phosphorylates STAT. Finally, STAT dimerizes and translo-cates into the nucleus regulating the expression of genes involved in cancer progres-sion [17,24–27]”. (Section Introduction, lines 78-83).
- 4. Line 55: Provide examples of how in silico drug repurposing has successfully identified anticancer agents in other malignancies.
ANSWER: Thanks for the suggestion. In the revised version of the manuscript, we added examples of how in silico drug repurposing helped to identify anticancer agents as reported in the cited literature, i.e. “The work by Melge et al. (2019) reports how disulfiram, an FDA-approved drug for al-coholism, was repurposed as a promising anticancer agent due to its ability to inhibit proteasomal activity and disrupt tumour cell metabolism. This discovery was sup-ported by molecular docking studies, which demonstrated the strong binding affinity of disulfiram’s active metabolites to cancer-related targets such as proteasome subu-nits. Another example presented is ivermectin, originally approved as an antiparasitic agent, which exhibited potential as an inhibitor of nuclear transport proteins in cancer cells. Through virtual screening and binding affinity assessments, ivermectin was shown to interfere with key pathways involved in tumorigenesis [31]”. (Section Introduction, lines 115-124), relating to the additional reference #31.
Discuss also the microenvironment of cancer cells, how they can affect immune cells, recent studies in breast cancer microenvironment should be mentioned, such as “Identification of the novel exhausted T cell CD8 + markers in breast cancer, 2024”
ANSWER: Thanks for the suggestion. In the Introduction section of the revised version of the manuscript, we have better described TME including immune cell activity with the additional sentences “In this context, immune surveillance is responsible for recognizing and eliminating the vast majority of cancer cells. Nevertheless, cancer cells employ various strategies to circumvent this vigilance, an ability recognized as one of the hallmarks of cancer [7]. The impact of the immune environment on cancers is particularly relevant in TNBC as evidenced by more recent works [8–10]” (lines 61-65). References #6-10 were added in order to address this issue.
- Line 70: Highlight gaps in current TNBC therapies that the study aims to address, emphasizing the lack of effective targeted treatments.
ANSWER: Thanks for the suggestion. We have included this topic in the revised version of the manuscript in the Introduction section, by adding the following sentences “To date, locoregional (surgery and radiotherapy) and systemic (chemotherapy, endocrine, and biological therapy) approaches represent the main therapeutic options for the treatment of BC. However, due to the histological heterogeneity and complex molecular profile of the tumour, patients frequently show different responses to the above treatments suggesting that new insights are needed as a step toward precision medicine [5]” (lines 44-49); “Due to these molecular features, the identification of new prognostic factors and pharmacological tools are needed to improve TNBC management” (lines 53-55); “In this study, we aimed to identify selective bioactive ligands able to influence IL-20RA activity. Therefore, IL-20RA may represent a new therapeutic target for the management of TNBC unresponsive to current treatments [34,35]” (lines 135-137).
Materials and Methods
- Line 100: Include details on the software and parameters used for virtual screening, particularly for docking score calculations. Cite previous studies using docking to support the approach, such as “Identification of molecular targets of Hypericum perforatum in blood for major depressive disorder: a machine-learning pharmacological study, 2024,Chebulinic Acid isolated from Aqueous Extracts of Terminalia chebula Retz inhibits Helicobacter pylori infection by potential binding to cag A protein and regulating adhesion, 2024,Isoform-specific N-linked glycosylation of NaV channel α-subunits alters β-subunit binding sites, 2025”
ANSWER: Thanks for the comment. In order to support our virtual screening and molecular docking, we followed protocols already used in our laboratory adding two more references #68 and #69.
- Line 130: Explain how potential ligands were selected for experimental validation, focusing on their predicted binding affinities and structural properties.
ANSWER: Thanks for the comment. As suggested, we gave further explanations for selecting these molecules in Results section: “Additionally, the selection of compounds was guided by a thorough review of the literature, which highlighted their mechanisms of action and their therapy indications, as shown in Table 1 and in Figure S1, Figure S2, and Figure S3. Based on this selection process, four molecules were identified: Goserelin, Triptorelin, Ritonavir, and Fenoterol. Notably, Goserelin [37,41,42] and Triptorelin [37,43,44] were selected due to their established approval as anticancer agents, underscoring their clinical relevance. Ritonavir [37,39,40] was included based on extensive documentation in the literature highlighting its in vitro antitumor activity. Fenoterol [37,45], in contrast, stood out due to its significantly lower molecular weight compared to the other compounds, a distinct structural feature that suggests a potentially unique mechanism of action or interaction, warranting further investigation” (lines 246-257), and in Materials and Methods section: “Compounds from each BS were evaluated based on their docking scores, with a cutoff set at 2 kcal/mol below the best pose” (lines 495-496).
- Line 150: Specify the rationale behind the selection of MDA-MB-157 and MDA-MB-231 cell lines for in vitro experiments. Mention some previous studies using MDA-MB-231 for cell cycle analysis might help, such as “The Role of Transient Receptor Potential Melastatin 7 (TRPM7) in Cell Viability: A Potential Target to Suppress Breast Cancer Cell Cycle, 2020,Effects of local anesthetics on breast cancer cell viability and migration, 2018”
ANSWER: Thanks for the comment. We added this information in the revised version of the manuscript, i.e. “MDA-MB-157 and MDA-MB-231, in vitro models of human TNBC widely used for their high ability of growth and progression” (Section Materials and Methods, lines 504-505).
Results
- Line 200: Discuss the biological relevance of the identified binding sites on IL-20RA, including their conservation across species.
ANSWER: Thanks for the comment. As suggested, we discussed the relevance of the binding sites on IL-20RA, including information about its phylogenetic evolution across species, i.e. “Furthermore, Zeng et al. (2024) have shown that IL-26, IL-10R2, and IL-20R1 share conserved structural motifs and functional domains between teleosts and higher vertebrates, including humans. Phylogenetic analyses demonstrate the clustering of these interleukins and their receptors with homologues in zebrafish and mammals, underscoring their ancient evolutionary lineage and functional significance [38]” (Section Results, lines 175-179).
- Line 240: Provide additional statistical analysis or graphical representation of docking scores for the top-ranked ligands.
ANSWER: Thanks for the comment. Apart from docking scores which were already present, further information, such as Molecular Weight, logP, Molecular target and Therapeutic application, were added in Table 1, Figure S1, Figure S2 and Figure S3.
- Line 280: Elaborate on the observed differences in antiproliferative effects between Ritonavir and the other tested ligands.
ANSWER: Thanks for the comment. In the revised version of the manuscript, we added this sentence: “At variance, for the other ligands the antiproliferative effect did not appear to be linear and dose and time-dependent (Figure S4)” (Section Results, lines 310-312).
- Line 320: Highlight potential off-target effects of Ritonavir and how these were mitigated in the study design.
ANSWER: Thanks for the comment. Toxicity studies in in vitro and in xenograft models of breast cancer have been included in Results section, i.e. “Moreover, this IC50 value corresponds to the one previously reported in other BC studies, in which ritonavir tumour inhibitory dose was well tolerated with acceptable toxicity in xenografts mice [46,47]” (Lines 317-320).
Discussion
- Line 400: Discuss the potential for Ritonavir to overcome drug resistance in TNBC, referencing its known effects on P-glycoprotein inhibition. Suggest potential future study for drug resistance target discovery, such as CRISPR screening, and how these can help, suggest to refer to “CRISPR screening and cell line IC50 data reveal novel key genes for trametinib resistance, 2025”
ANSWER: Thanks for the comment. In the revised version of the manuscript, we highlighted the potential application of Ritonavir to overcome drug resistance in TNBC by inhibiting P-glycoprotein, by adding the following sentences: “Ritonavir’s potential for cancer treatment is further supported by its ability to inhibit P-glycoprotein expression and activity, a major mediator of multidrug resistance (MDR) in cancer cells. In addition, it inhibits BC-resistant protein (BCPR) efflux, enhancing intracellular drug concentration [57,58]. These actions improve the accumulation of chemotherapeutic agents within tumour cells, making Ritonavir a valuable candidate for overcoming drug resistance [55,56]” (Section Discussion, lines 423-428).
Regarding CRISPR)/Cas9 technology, we would like to thank you for suggesting this kind of innovative study, since we think that it represents a very useful tool to discover genes associated with drug resistance and we will keep it in mind for future studies.
- Line 450: Address the limitations of using 2D in vitro models and propose follow-up studies involving 3D cultures or in vivo validation. Suggest future studies that could validate these findings in patient-derived xenograft models or larger cohorts. Previous studies using xenograft models of cancer should be mentioned, such as “Comparing volatile and intravenous anesthetics in a mouse model of breast cancer metastasis, 2018”
ANSWER: Thanks for the suggestion. We discussed this point by adding the following sentences: “Further studies in in vivo models will be needed in order to study pharmacological aspects strictly related to Ritonavir’s anticancer application, including the management of TNBC and other tumours characterized by high levels of IL-20RA expression” (Section Discussion, lines 449-452).
- Line 480: Explore the broader applicability of IL-20RA inhibitors in other cancers characterized by high IL-20RA expression.
ANSWER: Thanks. We discussed this point together with the previous suggestion in the final sentence of the Discussion section, i.e “Further studies in in vivo models will be needed in order to study pharmacological aspects strictly related to Ritonavir’s anticancer application, including the management of TNBC and other tumours characterized by high levels of IL-20RA expression” (lines 450-452).
Add additional discussion: Studies suggested that anesthetics during surgery treatment can impact cancer, reported in a series of work by Prof Lin’s group: “Effects of local anesthetics on breast cancer cell viability and migration, 2018,Effects of local anesthetics on cancer cells, 2020,Effect of Propofol on breast Cancer cell, the immune system, and patient outcome, 2018,Lidocaine Suppresses Viability and Migration of Human Breast Cancer Cells: TRPM7 as A Target for Some Breast Cancer Cell Lines, 2021,The Potential Effect of General Anesthetics in Cancer Surgery: Meta-Analysis of Postoperative Metastasis and Inflammatory Cytokines, 2023,Potential Therapeutic Application of Local Anesthetics in Cancer Treatment, 2022” These should be emphasized and discuss if the anesthetics impact involved in the mechanisms discussed in this study
ANSWER: Thanks. As suggested, we included this issue in the Discussion section, i.e. “Noteworthy, extensive research reported the ability of general anaesthetics to influ-ence the secretion of inflammatory cytokines such as IL-6, IL-10, and TNF-α in TME of BC. This, in turn, could heighten the risk of the development of lung cancer metastases after surgery. For example, in BC cases involving lung metastases, sevoflurane expo-sure during surgery has been observed to influence the lung TME through IL6/JAK/STAT3 signalling pathway regulation. Additional in vivo and in vitro studies are needed to clarify the mechanisms by which anaesthetics affect cancer cells and the immune system in TME [49,50]” (lines 383-390).
Figures and Tables
- Figure 2: Include annotations or labels on binding site visualizations to improve reader comprehension.
ANSWER: Done.
- Figure 5: Present raw data for cell viability assays in a supplementary table to ensure reproducibility.
ANSWER: Thanks. The requested data are reported in the new Table S9.
- Table 1: Add additional information about the chemical structures and properties of the four ligands tested.
ANSWER: Thanks. As suggested, we added further information such as Molecular Weight, logP, Molecular Target and therapeutic application in Table 1, Figure S1, Figure S2 and Figure S3.
Reviewer 2 Report
Comments and Suggestions for Authors
The authors used DrugBank to search for new compounds which could be used for the treatment of triple negative breast cancer, the most aggressive subtype of breast cancer without targeted therapy available. They focused on interleukin 20 receptor which regulates tumor microenvironment. The in silico methodology allowed to search for molecular mechanisms in drug-target interactions. Virtual screening allowed them to find four compounds for further molecular analysis.
The authors also tested their antitumor activity on two breast cancer cell lines. Only one of them, Ritonavir, reduced the proliferation of cancer cells in in vitro experiments. The manuscript is interesting showing that the silico analysis can hasten the discovery of new drugs for cancer treatment.
Author Response
Review Report Form #2
Comments and Suggestions for Authors
The authors used DrugBank to search for new compounds which could be used for the treatment of triple negative breast cancer, the most aggressive subtype of breast cancer without targeted therapy available. They focused on interleukin 20 receptor which regulates tumor microenvironment. The in silico methodology allowed to search for molecular mechanisms in drug-target interactions. Virtual screening allowed them to find four compounds for further molecular analysis.
The authors also tested their antitumor activity on two breast cancer cell lines. Only one of them, Ritonavir, reduced the proliferation of cancer cells in in vitro experiments. The manuscript is interesting showing that the silico analysis can hasten the discovery of new drugs for cancer treatment.
ANSWER: We thank the reviewer for his/her revision of the manuscript.
Reviewer 3 Report
Comments and Suggestions for Authors
I consider the article entitled “Rational identification of Ritonavir as IL-20 receptor A ligand endowed with antiproliferative properties”, to be very interesting and well written in general terms. However, some minor corrections need to be made. My observations and comments are as follows:
The authors should improve the wording of the title, as the paper is focused on mammary gland tumor cells, however, the title of the manuscript does not mention anything about breast cancer or MDA cells.
It seems to me that the Abstract and Introduction of the article do not present a clearly worded objective.
It seems to me that the following paragraph (lines 148-151) does not match what is described in the legend of Figure 3: “Additional grids were generated for the α-subunit using the same coordinates and dimensions as BS-A, but in one case considering only IL-20 (referred to as BS-B) and in the other considering only the α-subunit (referred to as BS-C).”
Lines 200-201: I believe that in this part of the document it would be interesting for the authors to briefly indicate the reason or motive for selecting four molecules.
Line 204: delete one of the words “and” that is repeated.
The image in Figure 4 does not indicate which image is (a), which is (b) and which is (c).
The caption (lines 234-235) of Figure 4 mentions the following: “π-cation and hydrogen bonding interactions are shown as green and yellow dashed lines, respectively”, however, this is not clearly observed in the images of that figure. The quality and/or size of the images in Figure 4 should be improved.
In section 4.7. Cell culture, proliferation and cell cycle assay, it is important that the authors indicate the reason for using two cell lines: MDA-MB-157 and MDA-MB-231.
In the legend of Figure 4, the authors should describe that CTR stands for Control and that it represents the behavior of tumor cells not treated with Ritonavir.
Line 431: the following expression should be corrected: 3.5x103 or 4x103. It should be 3.5x103 or 4x103.
Figure S4 does not include the results of Ritonavir, it would be necessary to include it to compare the activity of the 4 compounds with the same type of graphs.
In Figure 5, I consider that the quality of the images in section (c) should be improved; practically what is in blue color is not noticeable. In addition, it is not indicated, neither in the image nor in the legend of Figure 5, which are the units of measurement of the numbers placed in the tables of section (c).
Lines 273 and 277: correct the data “(200ng/m)” to “(200 ng/mL)”. This should also be corrected in the image in Figure 6.
The quality of the image in Figure 7b should be improved.
In Figure 7a the term “KDa” should be corrected. The correct is kDa.
Line 302: the term Kcal/mol is incorrect. Correct is kcal/mol.
Author Response
Review Report Form #3
Comments and Suggestions for Authors
I consider the article entitled “Rational identification of Ritonavir as IL-20 receptor A ligand endowed with antiproliferative properties”, to be very interesting and well written in general terms. However, some minor corrections need to be made. My observations and comments are as follows:
- The authors should improve the wording of the title, as the paper is focused on mammary gland tumor cells, however, the title of the manuscript does not mention anything about breast cancer or MDA cells.
ANSWER: Thanks for the suggestion. In the revised version of the manuscript the title has been changed adding “in breast cancer cells” as suggested.
- It seems to me that the Abstract and Introduction of the article do not present a clearly worded objective.
ANSWER: Thanks for the comment. In the revised version of the manuscript we clarify the aim of our study, by adding the following sentences: “Recent studies have highlighted the involvement of IL-20 receptor subunit alpha (IL-20RA) signalling in several cancers, including BC, in which IL-20RA is highly ex-pressed correlating with poor prognosis and influencing tumoural characteristics such as proliferation, cell death, invasiveness and TME activity. Therefore, elucidating the role of the IL-20RA signalling pathway could form the basis for developing new therapeutic strategies. This study aimed to identify selective bioactive ligands able to affect IL-20RA activity” (Section Abstract, lines 22-27); and in the Introduction section, “In this study, we aimed to identify selective bioactive ligands able to influence IL-20RA activity. Therefore, IL-20RA may represent a new therapeutic target for the management of TNBC unresponsive to current treatments [34,35]” (lines 135-137).
- It seems to me that the following paragraph (lines 148-151) does not match what is described in the legend of Figure 3: “Additional grids were generated for the α-subunit using the same coordinates and dimensions as BS-A, but in one case considering only IL-20 (referred to as BS-B) and in the other considering only the α-subunit (referred to as BS-C).”
ANSWER: Thanks for the comment. We specified the binding sites characteristics by adding different sentences along the manuscript: “Among the eight identified binding sites, three were selected for further investigation: two located at the interface between the IL-20 and the α-subunit, referred to as Binding Site-1 (BS-1), and one at the interface between IL-20 and the β-subunit, referred to as Binding Site-2 (BS-2 ) consistent with previous literature [16] (Figure 2). In fact, in the study by Logsdon et al. (2012) two critical binding sites, termed Site 1 and Site 2, were identified on the IL-20RA receptor. In particular, they described Site 1 as comprising two contact surfaces (Site 1a and Site 1b, which correspond to the two binding sites that we named BS-1). Site 1 is primarily responsible for the high-affinity interaction with IL-20, facilitating the initial cytokine-receptor engagement. Site 2 contributes to receptor dimerization and subsequent signal transduction. The structural integrity and functional relevance of these sites are crucial for IL-20-mediated signalling pathways [16]” (Section Results, lines 164-174); “Thus, taken together, BS-A, BS-B and BS-C are related to BS-1, while BS-D, BS-E and BS-F are related to BS-2.” (Section Results, lines 195-197). Figure 2 and its legend were modified in order to highlight BS-1 and BS-2 positions.
- Lines 200-201: I believe that in this part of the document it would be interesting for the authors to briefly indicate the reason or motive for selecting four molecules.
ANSWER: Thanks for the comment. As suggested, we gave further explanations for selecting these molecules, in Results section, i.e. “Additionally, the selection of compounds was guided by a thorough review of the literature, which highlighted their mechanisms of action and their therapy indications, as shown in Table 1 and in Figure S1, Figure S2, and Figure S3. Based on this selection process, four molecules were identified: Goserelin, Triptorelin, Ritonavir, and Fenoterol. Notably, Goserelin [37,41,42] and Triptorelin [37,43,44] were selected due to their established approval as anticancer agents, underscoring their clinical relevance. Ritonavir [37,39,40] was included based on extensive documentation in the literature highlighting its in vitro antitumor activity. Fenoterol [37,45], in contrast, stood out due to its significantly lower molecular weight compared to the other compounds, a distinct structural feature that suggests a potentially unique mechanism of action or interaction, warranting further investigation”(lines 245-256).
- Line 204: delete one of the words “and” that is repeated.
ANSWER: Done.
- The image in Figure 4 does not indicate which image is (a), which is (b) and which is (c). The caption (lines 234-235) of Figure 4 mentions the following: “π-cation and hydrogen bonding interactions are shown as green and yellow dashed lines, respectively”, however, this is not clearly observed in the images of that figure. The quality and/or size of the images in Figure 4 should be improved.
ANSWER: Thanks. Figure 4 has been improved both in quality, size and labels.
- In section 4.7. Cell culture, proliferation and cell cycle assay, it is important that the authors indicate the reason for using two cell lines: MDA-MB-157 and MDA-MB-231.
ANSWER: Thanks for the comment. We added this information in the revised version of the manuscript, i.e. “MDA-MB-157 and MDA-MB-231, in vitro models of human TNBC widely used for their high ability of growth and progression” (Section Material and Methods, lines 504-505).
- In the legend of Figure 4, the authors should describe that CTR stands for Control and that it represents the behavior of tumor cells not treated with Ritonavir.
ANSWER: Thanks. We supposed that the comment refers to Figure 5. As suggested, we clarified this information in the legend, by adding the following sentence: “Control, indicated as CTR, represents the untreated cells.”
- Line 431: the following expression should be corrected: 3.5x103 or 4x103. It should be 3.5x103 or 4x103.
ANSWER: Done.
- Figure S4 does not include the results of Ritonavir, it would be necessary to include it to compare the activity of the 4 compounds with the same type of graphs.
ANSWER: Thanks for the comment. To best compare the activity of the 4 compounds, the same type of graph used in Figure S4 is now shown in the new Figure 5a also for Ritonavir.
- In Figure 5, I consider that the quality of the images in section (c) should be improved; practically what is in blue color is not noticeable. In addition, it is not indicated, neither in the image nor in the legend of Figure 5, which are the units of measurement of the numbers placed in the tables of section (c).
ANSWER: Thanks. We improved the quality of the images. The numbers placed in the Tables correspond to the percentage of cells in the different cell cycle phases, as indicated by the sentence “The percentage of cells in each population is shown in the Tables below plots” in the legend of Figure 5.
- Lines 273 and 277: correct the data “(200ng/m)” to “(200 ng/mL)”. This should also be corrected in the image in Figure 6.
ANSWER: Done.
- The quality of the image in Figure 7b should be improved.
ANSWER: Done.
- In Figure 7a the term “KDa” should be corrected. The correct is kDa.
ANSWER: Done.
- Line 302: the term Kcal/mol is incorrect. Correct is kcal/mol.
ANSWER: Done.
Reviewer 4 Report
Comments and Suggestions for Authors
In the abstract, the purpose is not clear. Do the authors want to resolve problems associated with mechanisms, or just to affect IL-20RA activity?
Little information about mechanisms or specificity of these compounds was acheived from reading the manuscript.
Tables should be listed as to describe and compare pivotal parameters.
In vivo experiment should be performed to support applicability of these compounds.
Comments on the Quality of English Language
The language organization should be improved to clarify the central idea and significance of this study.
Author Response
Review Report Form #4
Comments and Suggestions for Authors
- In the abstract, the purpose is not clear. Do the authors want to resolve problems associated with mechanisms, or just to affect IL-20RA activity?
ANSWER: Thanks for the comment. In the revised version of the manuscript, we clarify the aim of our study. In particular, in Abstract section, we added the following sentences “Recent studies have highlighted the involvement of IL-20 receptor subunit alpha (IL-20RA) signalling in several cancers, including BC, in which IL-20RA is highly expressed correlating with poor prognosis and influencing tumoural characteristics such as proliferation, cell death, invasiveness and TME activity. Therefore, elucidating the role of the IL-20RA signalling pathway could form the basis for developing new therapeutic strategies. This study aimed to identify selective bioactive ligands able to affect IL-20RA activity” (lines 22-27); while in Introduction section we added the following sentence: “In this study, we aimed to identify selective bioactive ligands able to influence IL-20RA activity. Therefore, IL-20RA may represent a new therapeutic target for the management of TNBC unresponsive to current treatments [34,35]” (lines 135-137).
- Little information about mechanisms or specificity of these compounds was acheived from reading the manuscript.
ANSWER:As suggested, we gave further explanation for selecting these molecules, i.e. “Additionally, the selection of compounds was guided by a thorough review of the literature, which highlighted their mechanisms of action and their therapy indications, as shown in Table 1 and in Figure S1, Figure S2, and Figure S3. Based on this selection process, four molecules were identified: Goserelin, Triptorelin, Ritonavir, and Fenoterol. Notably, Goserelin [37,41,42] and Triptorelin [37,43,44] were selected due to their established approval as anticancer agents, underscoring their clinical relevance. Ritonavir [37,39,40] was included based on extensive documentation in the literature highlighting its in vitro antitumor activity. Fenoterol [37,45], in contrast, stood out due to its significantly lower molecular weight compared to the other compounds, a distinct structural feature that suggests a potentially unique mechanism of action or interaction, warranting further investigation” (Section Results, lines 244-255). Moreover, we added further information such as Molecular Weight, logP, Molecular Target and therapeutic application, in Table 1, Figure S1, Figure S2 and Figure S3.
- Tables should be listed as to describe and compare pivotal parameters.
ANSWER: Thanks for the comment. Since the docking scores of the selected molecules, correspond to the interactions with different BS, those parameters cannot be compared. For example, the considered docking scores for Triptorelin correspond to the interactions with BS-C and BS-E, while the docking scores of Goserelin correspond to BS-B and BS-F.
- In vivo experiment should be performed to support applicability of these compounds.
ANSWER: Thanks for the suggestion. The most interesting identified compound is Ritonavir, an anti-viral agent already in clinical use So, its in vivo safety tests have been yet performed, as reported in reference #40. As concerns its anti-cancer applicability, we agree with this referee that further experiments should be done and the results will be object of our next manuscript, more focused on the pharmacological aspects. Actually, we added the following sentence at the end of the Discussion section, i.e. “Further studies in in vivo models will be needed in order to study pharmacological as-pects strictly related to Ritonavir’s anticancer application, including the management of TNBC and other tumours characterized by high levels of IL-20RA expression.” (lines 449-452).
Comments on the Quality of English Language
- The language organization should be improved to clarify the central idea and significance of this study.
ANSWER: As suggested, English language has been revised throughout the manuscript.
Round 2
Reviewer 1 Report
Comments and Suggestions for Authors
ok